# Characterization of the Free and Membrane-Associated Fractions of the Thylakoid Lumen Proteome in *Arabidopsis thaliana*

**DOI:** 10.3390/ijms22158126

**Published:** 2021-07-29

**Authors:** Peter J. Gollan, Andrea Trotta, Azfar A. Bajwa, Ilaria Mancini, Eva-Mari Aro

**Affiliations:** 1Molecular Plant Biology, Department of Life Technologies, University of Turku, FIN-20014 Turku, Finland; andrea.trotta@ibbr.cnr.it (A.T.); azalba@utu.fi (A.A.B.); imancini1@uninsubria.it (I.M.); evaaro@utu.fi (E.-M.A.); 2Institute of Biosciences and Bioresources, National Research Council of Italy, Sesto Fiorentino, 50019 Firenze, Italy

**Keywords:** thylakoid lumen, protein, proteomics, photosynthesis

## Abstract

The thylakoid lumen houses proteins that are vital for photosynthetic electron transport, including water-splitting at photosystem (PS) II and shuttling of electrons from cytochrome *b_6_f* to PSI. Other lumen proteins maintain photosynthetic activity through biogenesis and turnover of PSII complexes. Although all lumen proteins are soluble, these known details have highlighted interactions of some lumen proteins with thylakoid membranes or thylakoid-intrinsic proteins. Meanwhile, the functional details of most lumen proteins, as well as their distribution between the soluble and membrane-associated lumen fractions, remain unknown. The current study isolated the soluble free lumen (FL) and membrane-associated lumen (MAL) fractions from *Arabidopsis thaliana*, and used gel- and mass spectrometry-based proteomics methods to analyze the contents of each proteome. These results identified 60 lumenal proteins, and clearly distinguished the difference between the FL and MAL proteomes. The most abundant proteins in the FL fraction were involved in PSII assembly and repair, while the MAL proteome was enriched in proteins that support the oxygen-evolving complex (OEC). Novel proteins, including a new PsbP domain-containing isoform, as well as several novel post-translational modifications and N-termini, are reported, and bi-dimensional separation of the lumen proteome identified several protein oligomers in the thylakoid lumen.

## 1. Introduction

Inside the plant chloroplast, a folded and continuous network of thylakoid membranes physically separates the soluble stroma (exterior) and lumen (interior) compartments. The membranes are embedded with large protein complexes that use light energy to oxidize water molecules in the lumen and transport electrons, ultimately generating reducing equivalents in the stroma. Electron transport along the thylakoid membrane develops an electrochemical gradient across the membrane that is harnessed by ATP synthase to produce ATP. Reducing equivalents and ATP power the reduction of CO_2_ in the chloroplast stroma into the stored chemical energy of carbohydrates. In this process, which is known as photosynthesis, the thylakoid lumen not only holds water for oxidation by photosystem (PS) II and accumulates protons for ATP synthesis, but also houses many protein components that are directly or indirectly involved in photosynthetic activity.

The major role of many lumenal proteins is to support electron transport through the membrane-embedded photosynthetic electron transport machinery powered by PSII and PSI. At the lumenal side, water-splitting and subsequent release of electrons, protons and molecular oxygen occur at the Mn_4_O_5_Ca cluster within the oxygen-evolving complex (OEC) of PSII. The OEC is supported by extrinsic lumenal PSII proteins PsbO, PsbP and PsbQ [1,2,3]. PSII ejects electrons to membrane-soluble plastoquinone (PQ), which carries them to the cytochrome (cyt) *b_6_f* complex. The plastocyanin (PC) protein relays electrons through the lumen from cyt *b_6_f* to PSI, via reversible reduction of its copper ion cofactor [4]. The NADH dehydrogenase-like (NDH) complex that spans the thylakoid membrane contains a multi-protein lumenal subcomplex that is required for its proton-coupled electron transport from NADPH to PQ (reviewed in [5,6]). Many lumenal proteins support the assembly and repair of membrane complexes involved in photosynthetic electron transport, especially of PSII that is frequently turned over in the light [7]. The Deg proteases are involved in degradation and removal of damaged D1 protein from the PSII core [8,9,10], while newly-inserted D1 precursors are processed to their functional form by lumenal C-terminal peptidases [11,12,13]. A number of other lumenal proteins have shown involvement in assembly and/or repair of PSII, PSI or cyt *b_6_f* complexes (reviewed in [14]).

Several proteomics studies have explored the thylakoid lumen of *Arabidopsis thaliana* (hereafter Arabidopsis), leading to experimental detection of approximately 55 lumenal proteins [15,16,17,18,19,20,21,22,23]. All known proteins localized to the Arabidopsis lumen are encoded by nuclear genes and transported across both the chloroplast envelope and thylakoid membrane courtesy of bipartite N-terminal peptides that are cleaved after translocation (reviewed in [24]). Identification of characteristic translocation peptide sequences has led to the prediction that many members of the thylakoid lumen proteome remain to be identified experimentally [15,16,17,20,25]. In this study we sought to use proteomics to determine the relative abundance of members of the lumenal proteome in the free soluble fraction of the lumen isolated from inverted thylakoids [15], as compared to the inner membrane-associated fraction liberated using urea and salt [26,27]. We report a vast range in the distribution and abundance of lumen proteins, as well as novel protein N-termini and post-translational modifications identified in many lumenal proteins.

## 2. Results

### 2.1. Comparison of the Free and Membrane-Associated Lumen Fraction

The thylakoid lumen proteome of Arabidopsis was obtained in two distinct fractions; the fraction of the proteome containing free, soluble proteins was collected as the supernatant after centrifugation of ruptured purified thylakoids, according to published lumen isolation methods [15,28]. After this ‘free lumen’ (FL) fraction was removed and the ruptured thylakoid membranes were thoroughly washed (see Methods), proteins that remained associated with the thylakoid membrane were detached by resuspending the membranes in 2.6 M urea and 200 mM NaCl [26,27]. After centrifugation, the ‘membrane-associated lumen’ (MAL) proteome was contained in the supernatant fraction. SDS-PAGE separation of the FL and MAL proteomes, alongside the total, lumen-free (FL removed) and stripped (both FL and MAL removed) thylakoid membranes revealed distinct differences in their respective protein compositions (Figure 1).

Immunoblot detection of selected marker proteins in the FL and MAL proteomes, and in the other thylakoid membrane fractions described above, showed that the two phases of the thylakoid lumen proteome had been effectively isolated (Figure 2). A lumenal FK506-binding protein (FKBP16-1) [20] was detected in the soluble FL fraction and not in the lumen-free membranes, demonstrating complete removal of the soluble fraction prior to MAL isolation. The PsbO protein has been found in both the soluble lumen and within the OEC protein complex associated with PSII at the inner thylakoid membrane [16,17,19]. Here, PsbO was detected by Western blotting in both the FL and MAL fractions, but was almost completely removed from the thylakoid by 2.6 M urea and 200 mM NaCl (Figure 2, lane 3). The photosystem I (PSI) subunit PsaN was virtually absent from the FL fraction, while a strong signal was detected in the MAL fraction. However, a portion of membrane-associated PsaN was not removed from the thylakoid membrane by 2.6 M urea and 200 mM NaCl, according to the PsaN signal detected in the ‘stripped’ membrane fraction (Figure 2, lane 3).

Four independent biological replicates of the FL and MAL fractions were analyzed using MS-based proteomics to identify members of the lumen proteome in each fraction. A list of verified or predicted thylakoid lumen proteins compiled from published studies [15,16,17,18,19,20,21,22,23] was used to identify lumen proteins in the current MS analysis, according to a threshold of at least ‘medium confidence’ (see Methods) with ≥2 peptide spectrum matches (PSMs) in ≥2 biological replicates. Using this approach, 60 members of the thylakoid lumen proteome were identified, while an additional 5 lumen proteins were detected with <2 PSMs and/or in <2 samples (Table 1). In addition, many other proteins, especially those associated with the stromal side of the thylakoid membrane, were unavoidable contaminants of both FL and MAL fractions, as demonstrated in previous studies [15,16,17]. In particular, ATP synthase subunits, ferredoxin-NADP^+^-oxidoreductase (FNR) and allene oxide synthase (AOS), as well as the Rubisco large and small subunits were found in relatively high abundance in the FL and MAL proteomes. For a list of all proteins identified in FL and MAL replicates, see Appendix A, respectively. Despite unavoidable contamination detected by highly sensitive MS-based proteomics, the FL and MAL fractions were highly enriched in *bona fide* lumenal proteins (Table 1, Appendix A). Quantification of all members of the lumen proteomes was beyond the scope of the current study; however, in order to explore the distribution of proteins between the FL and MAL proteomes, we approximated the abundance of proteins using the number of PSMs for each protein detected by MS [29]. PSMs were averaged across four replicate FL or MAL fractions. This analysis revealed a broad range in the relative abundance of lumenal proteins within each fraction, and further demonstrated the differing composition of the two lumen proteomes. 

The most abundant protein in the FL fraction (based on PSMs) was the PSII assembly component HCF136, which comprised approximately 15% of all lumen protein PSMs in that fraction (Table 1). Other abundant FL proteins included APX4, cyclophilin 38 (CYP38), PsbO1/PsbO2 and PsbP1, each of which contributed 5–10% of the FL proteome PSMs. Notably, PsbP1 abundance in the FL fraction varied considerably between the four biological replicates. The highly homologous PsbO1 and PsbO2 proteins could not be reliably distinguished by this analysis, and therefore the ratio between the PsbO isoforms was further analyzed in both the FL and MAL fractions using selected reaction monitoring (SRM) to quantify their unique peptides (Appendix A). This work demonstrated that the PsbO1:PsbO2 ratios were equivalent in both the FL and MAL fractions at approximately 6.5:3.5 (Figure 3). Unsurprisingly, the OEC proteins PsbP1 and PsbO1/O2 were the major components of the MAL proteome, each comprising 15–20% of that fraction. APX4, CYP38 and HCF136 were also abundant in the MAL fraction (5–10% each).

Enrichment of each protein in either the FL or MAL was approximated by comparing the relative abundance of each protein, calculated as a percentage of the total lumen protein PSMs in each sample and then averaged across four biological replicates. MAL/FL ratios of >1.5 were taken to indicate relative enrichment in the MAL, which applied to the OEC proteins, PsaN, the lipocalin LCNP (also called CHL) and several other proteins (see cells with double outline in Table 1). Conversely, MAL/FL ratios of <0.67 indicated relative FL enrichment, which applied to HCF136, each of the DegP and D1 proteases, most of the PsbP domain proteins and others (see cells with dashed outline in Table 1).

### 2.2. Novel Lumen Proteins

A PsbP-like protein identified in the current work to be among the 20 most highly represented proteins (based on PSMs) in the FL fraction corresponded to the TL19 protein (P82658; [16,18,23]), encoded by the gene AT3G63540. Translation of the putative AT3G63540 coding sequence (CDS) obtained from the Arabidopsis genome assembly (TAIR10) renders an incomplete PsbP domain protein lacking an N-terminal signal peptide. This may be related to the location of the gene at the extreme terminus of Arabidopsis chromosome 3 that may have impeded sequencing of the entire gene. BLAST searches of publicly available EST data were used to retrieve several full-length CDS expressed from the gene (e.g., EST sequence BAF10999), which were found to encode a complete PsbP domain and a putative N-terminal signal peptide for chloroplast and thylakoid transport (see Appendix A). The translated protein was homologous to 11 other PsbP domain-containing proteins occurring in Arabidopsis (reviewed in [30]), but constituted a unique member of the PsbP-like family that was therefore named PsbP domain 9 (PPD9). BLAST searching of sequenced genomes showed PPD9 to be conserved in all other photosynthetic species analyzed. Phylogenetic analysis of all PsbP domain proteins isolated from 11 photosynthetic species illustrated that the PPD9 amino acid sequence is most closely related to PPD4 and PPD6 (Figure 4). Notably, the thylakoid signal peptide of the PPD9 precursor was unlike all other Arabidopsis PsbP domain proteins, in that it did not possess the RR/KR motif typical of a twin arginine translocation (TAT) pathway substrate (reviewed in [24]). TAT substrate precursors were also absent from all other PPD9 orthologues analyzed, with the exception of the green algae *Chlamydomonas reinhardtii* and *Volvox carteri* (Appendix A), suggesting that the mechanism of PPD9 translocation into the thylakoid lumen is unique among plant PsbP domain proteins.

Other members of the thylakoid lumen proteome are reported here for the first time, having been previously inferred from gene and transcript sequence data. AT3G05410, AT2G01918 and AT4G19830 were predicted to encode the lumenal proteins PPD7 [31], PsbQ-like 3 (PQL3) [32] and FKBP17-1 [16,17], respectively, but were experimentally detected for the first time in the current study. All three have well-known lumenal paralogues [16,17,19,20,21,22]. Additionally, predictions that the AT2G26340 gene encodes a lumena protein [16,17,18,19] were verified for the first time in the current study, with the corresponding AT2G26340 protein (of unknown function) detected in both FL and MAL fractions (Table 1). Notably, the putative lumenal protein FKBP17-3 encoded by the gene AT1G73655 [33] was not detected in the current analysis and remains unreported at the protein level.

### 2.3. N-Termini

Lumenal proteins are translocated across the thylakoid membrane from the stromal side and then processed to their mature forms by proteolytic cleavage of the N-terminus of the precursor protein (reviewed in [24]). The N-termini of lumenal proteins therefore indicate the cleavage sites of the thylakoid processing peptidase (TPP; reviewed in [34]). Reprocessing of our MS data allowing for semi-tryptic cleavage enabled the identification of N-terminal peptides from over 65% of the proteins found in the lumen. Novel protein N-termini highlighted in Table 1 include those of newly identified proteins (described above) and proteins that have thus far escaped N-terminal analysis, including CHL, CYP26-2 and FKBP17-1. Additionally, several novel N-termini reported here differ from those that have been identified in previous proteomics analyses, such as for PsbQ-like 2 (PQL2), FKBP16-2 and the pentapeptide repeat protein (PRP) TL20.3 (see Table 1). MS analysis also identified novel instances of acetylation in some N-termini, including of PsbQ2, CYP20-2, CYP38 and FKBP17-2 (Table 1). The N-termini of CYP28 and FKBP17-2 detected by MS likely derived from the stromal precursors, based on close proximity to the full-length N-termini (positions 20 and 29/30, respectively, see Table 1). In both cases, secondary semi-tryptic N-termini downstream (CYP28 position 60, FKBP17-2 position 61; Table 1) may represent the N-termini of the mature, lumenal proteins. The same might be true for PPD6 (position 61), PsbP1 (position 70) and FKBP16-4 (position 92) (Table 1).

### 2.4. Protein Oligomer Complexes in the Lumen

For analysis of protein complexes, the FL fractions were run on clear native (CN)-PAGE that preserves potential native interactions between proteins. Individual lanes cut from CN-PAGE were loaded onto denaturing SDS-PAGE for separation of proteins according to their molecular mass. The second dimension SDS-PAGE gels were stained with SYPRO Ruby (Figure 5) and then silver, allowing protein spots of interest to be excised and the component proteins identified by MS (listed in Table 2). To integrate the information on the lumen proteome obtained by total soluble proteins analysis (see above), we focused on protein spots with low abundance. According to their migration behavior during the CN-PAGE analysis, many lumenal proteins appeared to participate in homo- or hetero-oligomeric complexes in the FL sample. In particular, PPD1, PPD4, PPD5, TLP18.3, APX4, FIKBP16-3 and TL15 migrated at a relatively high molecular weight in CN-PAGE (Figure 5). PsbO1/PsbO2 appeared to migrate in the native dimension together with a large protein spot that contained PsbP domain protein PPD2 and the PRP TL17, while the OEC protein PsbP1 was identified only at very low abundance (below cut-off) in the same spot. The major isoform of PC was detected in two abundant protein spots that appeared to migrate as free proteins in CN-PAGE.

### 2.5. Phosphorylated Proteins/Peptides in the Thylakoid Lumen

Phosphorylated proteins were detected by MS in protein spots excised from two-dimensional CN/SDS-PAGE gels that were stained with Pro-Q Diamond (Figure 5). This approach identified 25 unique phosphopeptides from 9 different lumenal proteins (Table 2), with several phosphorylation cases previously unreported. The current approach could not definitively determine PsbO2 phosphorylation, due to complete identity of two phosphopeptides between both isoforms (see Table 2). Phosphorylation of PsbO1/O2 has been shown [35,36]; however, the current study identified additional novel phosphorylation sites in PsbO1/O2. Similarly, CYP38, a known phosphoprotein [36,37], was shown in the current work to be heavily phosphorylated, with four novel sites reported here (Table 2). PPD2 and the major PC isoform are reported here as novel phosphoproteins, as are the PRP proteins TL15 and TL17, which were found to contain three and four phosphorylation sites, respectively (Table 2). Additionally, the uncharacterised proteins TLP15 and AT2G23670 were found to be novel phosphoproteins.

**Table 2 ijms-22-08126-t002:** Mass spectrometry-based identification of proteins isolated by bi-dimensional clear native PAGE, including phosphorylation sites.

Accession	Name	Phosphorylation Site(s) *	Phosphorylation Reported **
AT3G01480.1	CYP38	Y109: (pY)ALPIDNKAIR S159:(pS)IIVAGFAESK T291: (pS)DGFVVQTGDPEGPAEGFIDPSTEKTR T298: SDGFVVQ(pT)GDPEGPAEGFIDPSTEKTR T318: (pT)VPLEImVTGEK S380: E(pS)ELTPSNSNILDGR	T318 experimentally detected (PhosPhAt, study not detailed) S159 [36]
AT5G66570.1	PsbO1	T108: G(pT)GTANQCPTIDGGSETFSFKPGK T108/T110: G(pT)G(pT)ANQCPTIDGGSETFSFKPGK S221: QLDA(pS)GKPDSFTGK T251: GGS(pT)GYDNAVALPAGGRGDEEELVK T278 ***: N(pT)AASVGEITLK T296 ***: SKPE(pT)GEVIGVFESLQPSDTDLGAK T305 ***: SKPETGEVIGVFE(pS)LQPSDTDLGAK	S226, S281 on the same peptides as S221, T278, respectively [35] S221 [36]
AT3G50820.1	PsbO2	T277 ***: ENVKN(pT)AASVGEITLK T295 ***: SKPE(pT)GEVIGVFESLQPSDTDLGAK	
AT4G15510.1	PPD1	T183: QYL(pT)EF(oM)STR	S28 [35]
AT2G28605.1	PPD2	S126: IK(pS)LDQFGSPQFVADK	S126 predicted (PhosPhAt)
AT1G77090.1	PPD4	no phosphopeptide detected	
AT5G11450.1	PPD5	no phosphopeptide detected	T283 [38] Y250 predicted (PhosPhAt)
AT1G20340.1	PC major	Y103: NNAG(pY)PHNVVFDEDEIPSGVDVAK	Y103 predicted (PhosPhAt)
AT4G18370.1	DegP5	no phosphopeptide detected	
AT4G09010.1	APX4 (TLP29)	no phosphopeptide detected	S155 [39]
AT2G44920.2	TL15	S99: GQDL(pS)GKDFSGQTLIR T164: VNL(pT)NANLEGATVTGNTSFK T190: GSNITGADF(pT)DVPLRDDQR	S99 predicted (PhosPhAt)
AT5G53490.3	TL17	T114: AFVGN(pT)IGQANGVYDKPLDLR T161: FDGAD(oM)(pT)EVV(oM)SK Y169: A(pY)AVEASFK T181: GVNF(pT)NAVIDR	T114 predicted (PhosPhAt)
AT2G43560.1	FKBP16-3	no phosphopeptide detected	
AT1G05385.1	Psb27-2 (LPA19)	no phosphopeptide detected	
AT5G52970.1	TLP15	Y129: VLAQN(pY)PVTPGLAIK	
AT4G02530.1	TLP16	no phosphopeptide detected	
AT1G54780.1	TLP18.3	no phosphopeptide detected	
AT2G23670.1	AT2G23670	T100: EGFE(pT)AEKGVDAAEK	T100 predicted (PhosPhAt)

* Phosphorylated residue indicated by (pX), oxidised methionine indicated with (oM). ** Phosphopeptides reported in literature or predicted on PhosPhAt database [37]; *** These sites are on peptides that are identical between PsbO1 and PsbO2. An image and complete list of protein spots isolated from CN/SDS-PAGE are provided in Appendix A and Appendix A.

## 3. Discussion

The known functions of proteins residing in the thylakoid lumen are mainly associated with supporting photosynthetic electron transport in the thylakoid membrane, either through promoting the activity of membrane-embedded protein complexes or by facilitating assembly, stability or turnover of the photosynthetic proteins (reviewed in [14]). Although limited, these details indicate that many members of the lumen proteome undergo functional association with the thylakoid membrane through direct or indirect interactions with membrane-intrinsic proteins. The distribution of the lumen proteome between the soluble and membrane fractions is therefore of interest. In the current work, we separated the thylakoid lumen proteome into two fractions; the soluble, free lumen (FL) fraction that was released upon inversion of thylakoid membranes [15,28], and the membrane-associated lumen (MAL) fraction that was dissociated from inverted thylakoids by interfering with both the electrostatic protein-protein interactions, using NaCl, and with the hydrogen bonding network, using the chaotropic agent urea [26,27]. Although it was not possible to completely avoid contaminating proteins from non-lumen sources, this method delivered lumen-enriched fractions which were combined with MS-based proteomics to identify 60 proteins in the FL proteome and 58 in the MAL proteome (Table 1). Furthermore, we used the spectral counts (PSMs) derived from MS to approximate the abundance of each protein within the two proteomes.

### 3.1. The Soluble and Membrane-Associated Fractions of the Thylakoid Lumen Are Distinct

Despite thorough membrane washing before MAL isolation, many proteins found in FL were also detected by MS at low abundances in the MAL fraction. This may have been due to the presence of a small amount of FL contained within membrane vesicles, even after two rounds of rupture by Yeda press, leading to minor contamination of the MAL that was detectible by the highly-sensitive MS. For example, over 2 PSMs of the lumen marker FKBP16-1 were detected in MAL replicates, while FKBP16-1 was not detected by western blot in the MAL fraction (Figure 2). Nonetheless, higher relative abundance of many proteins in the MAL fraction than in the FL (Table 1, Figure 1) suggested genuine distribution of some proteins to both proteomes and enrichment in the MAL proteome. According to the current study, the OEC proteins were found in both fractions and were significantly enriched (>2-fold) in the MAL proteome, compared to the FL. The PSI subunit PsaN, the chloroplast lipocalin LCNP, the PRP TL20.3, as well as FKBPs 16-4 and 17-2, were also MAL-enriched. The existence of both soluble and membrane-associated pools of the OEC proteins is well-established [3,40], and this was reiterated here with PsbO1/O2 and PsbP1 being major and stoichiometrically equivalent proteins of both FL and MAL fractions (Table 1), although substantial variability in PsbP1 abundance between the FL replicates suggested that the protein may not be a consistent member of the soluble fraction. In SDS-PAGE separations of the lumen proteomes, a major 22 kDa band corresponding to PsbP1 was apparent in the MAL, but not in the FL fraction (Figure 1). PsbP1 was also not prominent after 2D separation of the FL in the current work (Figure 5), unlike in previous studies [16,22], but was found below the cut-off value (>5 PSMs) in spots 7 and 8 (see Appendix A and Appendix A). Notably, PsbQ1 and PsbQ2 were far less abundant than the other OEC proteins in both fractions, but were nonetheless prominent within the MAL proteome. It was not surprising to find PsaN, a lumenal subunit of the PSI complex required for electron transport from PC [41], to be significantly enriched in the MAL. Enrichment of LCNP in the MAL fraction also corresponds with its role in managing thylakoid lipid peroxidation during abiotic stress [42,43], although the related lumenal protein violaxanthin de-epoxidase (VDE), responsible for de-epoxidation of membrane-intrinsic carotenoids during high light stress (reviewed in [44]), was instead enriched in the FL fraction (Table 1). The plants used in the current study were harvested from ‘non-stress’ conditions, which suggests that LCNP may be permanently associated with the inner thylakoid membrane, while VDE is only transiently membrane-associated during stress conditions. The roles of other MAL-enriched proteins TL20.3, FKBP16-4 and FKBP17-2 are currently unknown, but their association with the thylakoid membrane suggests a functional interaction with membrane-bound protein partners. Importantly, the current study shows that PsaN, LCNP and other MAL-enriched proteins that are isolated using traditional lumen (FL) isolation methods represent only a minor fraction of those lumenal proteins.

### 3.2. A Major Role of the Lumen Proteome Is PSII Assembly and Maintenance

During photosynthesis, the D1 protein in the core of PSII is frequently damaged and replaced with newly-synthesised D1 during PSII repair (reviewed in [45]). The prominence of lumenal proteins involved with assembly and turnover of PSII shown here illustrates that this process is a primary concern of the lumen proteome in mature leaves. The PSII assembly chaperone HCF136 was the most abundant protein in the FL fraction, highlighting the importance of its role in stabilising newly-synthesised D1 precursor protein from the lumenal side during PSII biogenesis and repair [46,47]. Also implicated in PSII assembly and repair, the cyclophilin CYP38 [48,49] and the PsbP-like PPL1 [50] were also highly abundant in the FL fraction. Three lumenal DegP proteases carry out degradation of damaged D1 from the lumenal side prior to removal. The current results showed DegP1, which is activated by homo-hexamerisation, to be the most abundant protease in the FL fraction, as was previously identified [51]. Due to similar number of peptides and protein coverage between DegP8 and DegP5, it was possible to estimate that DegP8 was approximately four times more abundant than DegP5. As DegP8 and DegP5 together form a hetero-hexamer [9], this result suggests unequal contributions to the protease complex from the two isoforms in FL. Prior to reassembly of repaired, functional PSII complexes, the lumenal extension of a newly-inserted D1 precursor is cleaved by the C-terminal-processing peptidase CtpA (reviewed in [52]), of which two isoforms were found to be abundant in the FL fraction (Table 1). These results underline the major function of the thylakoid lumen in maintaining the activity of PSII, and indicate that other prominent lumenal proteins with unknown roles may also be involved with PSII assembly, stability and/or repair. In particular, CYP37 and the ‘inactive’ ascorbate peroxidase APX4 [53,54] present strong cases for potential PSII chaperones.

### 3.3. Protein Families Populate the Thylakoid Lumen

According to the current results, around 20% of the FL proteome (as approximated by spectral counts) consisted of PsbP and other members of the PsbP-like protein family, with PsbP-like proteins comprising over 13%. While the role of PsbP in interacting with PSII to maintain the Mn_4_O_5_Ca cluster in the OEC is well known (reviewed in [55,56]), the functions of the PsbP homologues are far less clear. PPL1, which is involved in PSII assembly [50], was found here to be highly abundant in the FL fraction (Table 1), as were PPD1, required for PSI assembly [57] and PPD2. The discovery here of a new PsbP domain protein PPD9 that is conserved across photosynthetic eukaryotes does not enlighten the function(s) of this protein family, but it does underscore the importance of the family in the lumen. The immunophilins, comprising the cyclophilin (CYP) and FK506-binding protein (FKBP) families, were also populous in the thylakoid lumen, contributing almost 20% of the FL proteome (as approximated by spectral counts). Defined by their theoretical capacity to rotate peptidyl-prolyl bonds, most plant immunophilins instead appear to operate as molecular chaperones with specific client proteins (reviewed in [58,59]). A few lumenal immunophilins have been implicated in PSII complex assembly [48,60] or formation of the lumenal NDH subcomplex [61], while the roles of the majority await discovery.

### 3.4. Post-Translational Modifications in the Thylakoid Lumen

The current work provides strong evidence supporting post-translational modifications of lumen proteins, for which a clear role is still missing. Nine different lumen proteins were found to be phosphorylated, often with multiple phosphorylation sites detected in a single protein (Table 2), with a majority of the phosphopeptides reported here for the first time. While the function of lumenal protein phosphorylation is largely unknown, published results indicate that phosphorylation of PsbO in wheat inhibits interaction between that protein and the PSII complex, instead promoting PsbO degradation [62]. Conversely, phosphorylation was recently shown to stabilize the PPD5 protein [38]. Phosphorylation of PsbO1/O2, CYP38, and PPD1 and PPD2 detected here (Table 2) and elsewhere [35,36,37] suggests that phosphorylation may regulate the functional interactions, stability and degradation of many lumen proteins in Arabidopsis. The PRP proteins TL15 and TL17 were shown here to be multiply phosphorylated within the coiled, central region of each protein that is formed by the repeated pentatricopeptide sequences. This region of the PRP proteins is known to be especially stable [63], suggesting that phosphorylation events detected here might interrupt the stability and function of TL15 and TL17. Two prominent protein spots containing the same phosphorylated form of the major PC isoform indicated a migration shift in the CN-PAGE that was apparently unrelated to the phosphostatus of the protein, but may be evidence of a pool of PC homo-oligomer in the FL fraction. A lumenal protein kinase responsible for protein phosphorylation in the lumen has long been speculated (reviewed in [64]), but has not been convincingly demonstrated. The current work also failed to identify any possible lumenal kinase, suggesting that lumenal proteins are more likely phosphorylated in the chloroplast stroma prior to translocation. On the other hand, the protein phosphatase TLP18.3, which was found to be a low-abundance lumen protein (Table 1, Figure 5), may provide a means of protein dephosphorylation for the activation/deactivation of phosphorylated proteins [65].

Notably, most proteins found in this study to be N-terminally acetylated (Table 1) are functionally associated with the thylakoid membrane, including PsbQ2, CYP38 and CYP20-2. In addition, N-terminal acetylation of PsaN, part of membrane-bound PSI, was previously identified [21], although not in the current study. Acetylation of the N-terminal amino acid of mature lumen proteins indicates the existence of both an N-terminal acetyl transferase and acetyl coenzyme A metabolites within the thylakoid lumen [66], perhaps at the lumenal face of the thylakoid, however these have not yet been discovered.

### 3.5. Concluding Remarks

The thylakoid lumen is vital for the assembly, function and maintenance of efficient photosynthetic electron transport, and several lumen proteins are known to be involved in these processes, but the functions of most lumen proteins remains elusive. The current work has identified differences in composition of the free lumen (FL) and the membrane-associated lumen (MAL) proteomes and the relative abundance of their constituents. The abundant proteins were only few, and most were involved in assembly and function of PSII, especially the OEC proteins, HCF136 and CYP38, although some highly abundant proteins such as APX4 and CYP37 remain to be functionally characterized. The majority of the lumen proteome (by number) comprised low-abundance proteins, such as most members of the large immunophilin and PsbP domain-containing families, the latter now including a newly-discovered member PPD9. Separate analysis of the FL and MAL fractions highlighted the distribution of PsbO isoforms in both fractions, and showed relative enrichment of PsbP and PsbQ isoforms, as well PsaN and LCNP, at the inner thylakoid membrane. A number of novel phosphorylation and N-terminal acetylation events detected in the current work strongly suggest regulation of several lumenal proteins by post-translational modifications that control their stability and/or distribution, although the modifying enzymes involved remain to be identified. This study has provided many new characteristics of the lumen proteome that will assist in developing a better understanding of this vital, yet mysterious compartment.

## 4. Materials and Methods

### 4.1. Plant Growth and Isolation of Thylakoid Lumen Fractions

*Arabidopsis thaliana* (L.) Heynh. Col-0 plants were grown for 6 weeks in a phytotron in short day conditions (8 h light/16 h darkness) under light intensity of 100 µmol photons m^−2^ s^−1^. Total vegetative tissues of approximately 50 plants were harvested with scissors and the free thylakoid lumen fraction was immediately isolated according to the method of [28]. All isolation steps were carried out under dim light in a cold room. All buffers used were ice cold. Plant tissue was ground in a large blender in ice-cold grinding buffer containing 330 mM sorbitol, 50 mM Hepes-KOH (pH 7.8), 10 mM KCl, 10 mM EDTA and 10 mM NaF, followed by filtration through Miracloth (MilliporeSigma, Burlington, MA, USA) mesh with a pore size of 20 µm. The filtrate was centrifuged in 50 mL polypropylene tubes at 1000× *g* at 4 °C for 1 min, the supernatant discarded and the pellet suspended using a soft brush in resuspension buffer containing 330 mM sorbitol, 20 mM Hepes-KOH (pH 7.8), 10 mM KCl, 2.5 mM EDTA, 5 mM NaCl and 10 mM NaF. After centrifugation at 1000× *g* at 4 °C for 1 min, the supernatant was discarded and the pellets resuspended in a minimal volume of the above buffer and combined. The suspension containing intact chloroplasts was diluted to a chlorophyll concentration of 0.2 mg ml^−1^ in shock buffer containing 10 mM Na_4_P_2_O_7_ (pH 7.8) and 10 mM NaF, and homogenized thoroughly with a glass homogenizer, before centrifugation in clean 50 mL polypropylene tubes at 7500× *g* at 4 °C for 5 min. After discarding the supernatant, the resulting thylakoid membrane pellet was washed once with shock buffer, twice with wash buffer containing 2 mM Tricine (pH 7.8) and 300 mM sucrose, and once with fragmentation buffer containing 30 mM NaH_2_PO_4_ (pH 7.8), 50 mM NaCl, 5 mM MgCl_2_ and 100 mM sucrose by repeated centrifugations at 7500× *g* at 4 °C for 5 min. After washing was completed, thylakoid pellets were resuspended in a minimal volume of fragmentation buffer containing 10 mM NaF and EDTA-free protease inhibitor cocktail (Merck, Darmstadt, Germany). The thylakoid solution was passed slowly through a cold Yeda press twice at N_2_ pressure of 10 MPa. The resulting solution of inverted membranes was centrifuged twice at 200,000× *g* at 2 °C for 60 min. The supernatant containing the ‘free lumen’ (FL) proteome was removed and stored at −80 °C. The membrane pellet was carefully washed by resuspension with a glass homogenizer in fragmentation buffer in order to remove any contaminating FL, followed by centrifugation at 200,000× *g* at 2 °C for 60 min. The pellet was then resuspended to 1.0 mg ml^−1^ chlorophyll in stripping buffer containing 2.6 M urea, 0.2 M NaCl, 50 mM CaCl_2_, 20 mM MES (pH 6.0) and 300 mM sucrose [27] using a glass homogenizer. The thylakoid suspension was incubated on ice for 60 min, and then centrifuged twice at 200,000× *g* at 2 °C for 60 min. The supernatant containing the ‘membrane-associated lumen’ (MAL) proteome was collected and stored at −80 °C. Protein concentrations were measured using a modified Lowry method [67].

### 4.2. Western Blotting

10–30 µg proteins were separated on SDS-PAGE containing 15% acrylamide and transferred to PVDF membranes that were blocked with 5% low fat milk and incubated with antibodies designed to recognise PsbO (Innovagen, Lund, Sweden), PsaN (AS06 109, Agrisera, Vännäs, Sweden) or FKBP16-1 (Agrisera). Antibody cross reactions were detected using secondary antibodies conjugated to horse radish peroxidase (Agrisera) and a chemiluminescent substrate (GE Healthcare, Madison, WI, USA).

### 4.3. Clear Native PAGE Separation and Protein Visualization

Bi-dimensional clear native (CN)-PAGE/SDS-PAGE was performed as in [29], with slight modifications. 50 µg of free lumen (FL) fractions were separated by 9.5–14% CN-PAGE in the first dimension, after which proteins were separated in second dimension by loading the CN-PAGE lane across the top of an SDS-PAGE gel containing 15% acrylamide and 6 M urea. The second-dimension gels were subsequently stained with ProQ Diamond (Invitrogen, Molecular Probes, Eugene, OR, USA) to visualize the phosphoproteins, and then with SYPRO Ruby to visualize all proteins, following manufacturer instructions (Invitrogen, Molecular Probes). Gels were then stained with silver according to the method described in [68] and protein spots of interest were excised from the gel and analyzed by mass spectrometry (MS), as described below.

### 4.4. Proteomics Methods

Lumen proteins were prepared for analysis in MS as in [69] with slight modifications. For analysis of total lumen proteomes, 20 µg of free lumen (FL) or membrane-associated lumen (MAL) proteins were loaded onto an SDS-PAGE gel containing 6% acrylamide and 6 M urea 6%, run approximately 0.5 cm and digested in-gel with 2 µg of trypsin. For analysis of protein spots isolated by bi-dimensional CN-PAGE, in-gel digestion of the protein spots was performed as in [29]. Peptides were separated in an Easy nLC 1000 HPLC system (Thermo Fisher Scientific, Waltham, MA, USA) using a 20 × 0.1 mm (inner diameter) precolumn, followed by a 150 mm × 75 mm (inner diameter) analytical column, both packed with 5 mm Reprosil C18-bonded silica (Dr Maisch GmbH, Ammerbuch-Entringen, Germany). Peptides were separated by an elution gradient of solvent A (water:acetonitrile (98:2 [*v*/*v*]) with 0.2% formic acid) and solvent B (water:acetonitrile 80:20 [*v*/*v*] with 0.2% formic acid) at a flow rate of 300 mL min^−1^. Gradients comprised 3% to 43% solvent B over 60 min (for total soluble proteins)/13 min (for gel spot samples), followed by 43% to 100% solvent B for 5 min/2 min, and finally 100% solvent B for 10 min/5 min.

The HPLC system was coupled to a Q-Exactive electrospray ionization-hybrid quadrupole-orbitrap mass spectrometer (Thermo Fisher Scientific), operating in positive mode. Data-dependent acquisition (DDA) of the generated MS/MS spectra was performed with higher-energy collisional dissociation (HCD) fragmentation, scan range set from 300 to 1800 *m*/*z* (MS1) and 200 to 2000 *m*/*z* (MS2), and up to 20 data-dependent MS/MS spectra acquired in each scan. Dynamic exclusion was set to 20 s and resolution was set to 140,000 (MS1) and 17,500 (MS2).

All acquired spectra were matched against a non-redundant Arabidopsis proteome database (TAIR10; https://www.arabidopsis.org/, (accessed on 27 February 2019)) supplemented with most common contaminants (35,502 entries in total), using Proteome Discoverer (PD, version 2.2) Software (Thermo Fisher Scientific) with an in-house Mascot (version 2.6.1; Matrix Science, Boston, MA, USA) search engine. The search parameters were set to monoisotopic mass, two missed cleavages allowed, precursor mass tolerance 10 ppm, fragment mass tolerance 0.02 Dalton, *m*/*z* ≥ 2. The variable modifications were oxidation of methionine, acetylation of protein N-terminus, phosphorylation of threonine, serine or tyrosine residues, while carbamidomethylation of cysteine was set as a fixed modification. In order to find potential N-terminal peptides of the mature lumen proteins, the same analysis was repeated with the semi-trypsin setting, which allows the identification of peptides with either N-terminal or C-terminal residues different from arginine or lysine. False discovery rate of the identified peptides was evaluated using the Percolator node in PD2.2 (for total soluble proteins) or Mascot score (for protein spots). The phosphorylations were validated using ptmRS node [70] in PD2.2. The MS proteomics data have been deposited to the ProteomeXchange Consortium via the PRIDE [71] partner repository with the dataset identifier PXD027229 and 10.6019/PXD027229.

Quantification of PsbO1 and PsbO2 isoforms was performed via selected reaction monitoring (SRM) as described in [69]. A protein band containing PsbO1 and PsbO2 was excised from SDS-PAGE and subjected to in-gel tryptic digestion and the PsbO1 and PsbO2 peptides were identified by MS as described above. The MS/MS spectra of unique (proteotypic) PsbO1 and PsbO2 peptides were used to generate a spectral library. The transitions used to detect proteotypic PsbO1 and PsbO2 peptides in SRM and the transitions used for quantification are listed in Appendix A. The PsbO1/PsbO2 ratio was calculated as the relative percentage of each isoform to the sum of the integrated peak area of the three most intense transitions of every proteotypic PsbO1 or PsbO2 peptide. The refined dataset of the SRM quantification is available in Panorama repository (https://panoramaweb.org/fvhab3.url) and the corresponding mass spectrometry raw files are available at the ProteomeXchange dataset PXD027514 (http://proteomecentral.proteomexchange.org/cgi/GetDataset?ID=PXD027514).

### 4.5. Bioinformatics Methods

Phylogenetic analysis was performed using Molecular Evolutionary Genetics Analysis (MEGA) version 6.0 [71]. Protein sequences were acquired from the Phytozome 12 plant genomics database (https://phytozome.jgi.doe.gov/pz/portal.html#). Protein details, including molecular weight and N-termini, were accessed from UniProt (www.uniprot.org/). Protein phosphorylation predictions were acquired from the PhosPhAt database version 4.0 (http://phosphat.uni-hohenheim.de/phosphat.html) [37].

## Figures and Tables

**Figure 1 ijms-22-08126-f001:**
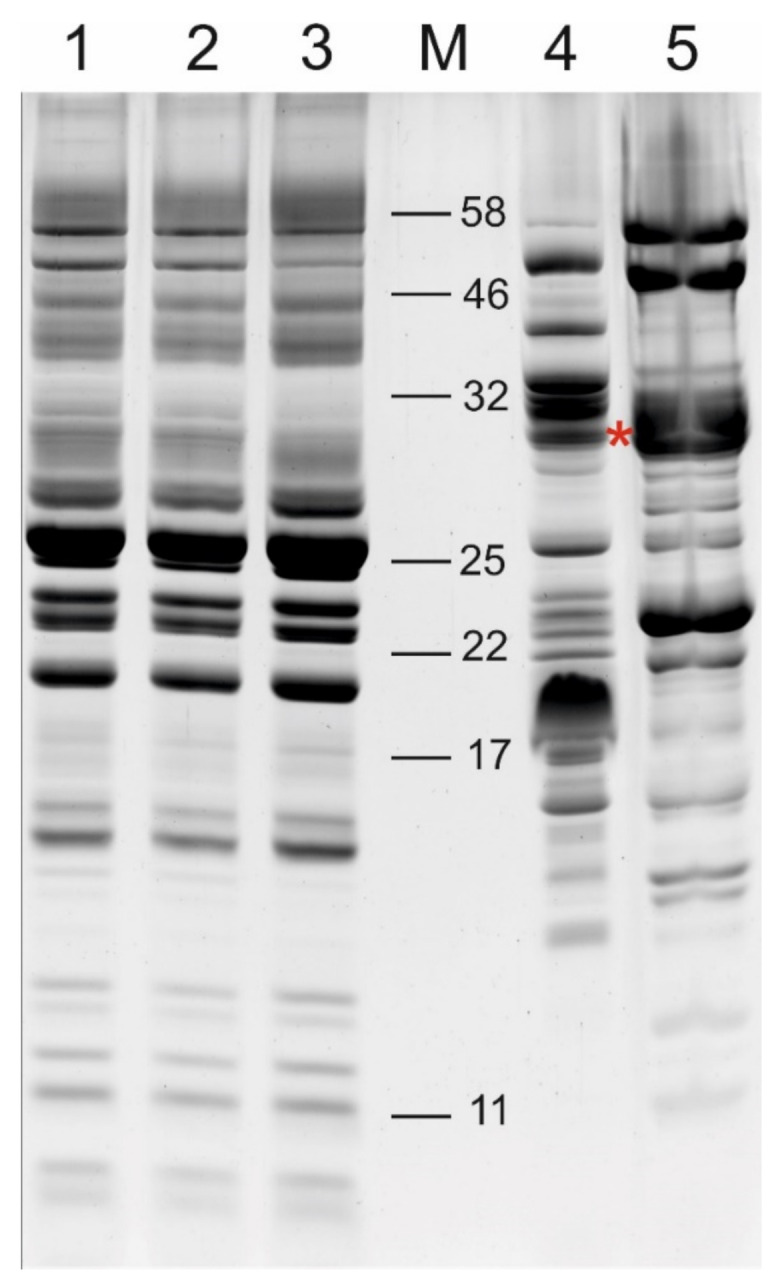
Thylakoid membrane and lumen fractions isolated during lumen preparations. SDS-PAGE separation of proteins stained with SYPRO Ruby. 1—total thylakoids; 2—thylakoid membranes after removal of the free lumen fraction; 3—thylakoid membranes after removal of the membrane-associated lumen fraction; 4—free lumen; 5—membrane-associated lumen; M—molecular weight marker. Lanes 1–3 contain 20 µg protein, lanes 4–5 contain 100 µg protein. Asterisk (*) indicates position of protein bands excised from FL and MAL fractions for PsbO1/PsbO2 analysis by selected reaction monitoring (SRM).

**Figure 2 ijms-22-08126-f002:**
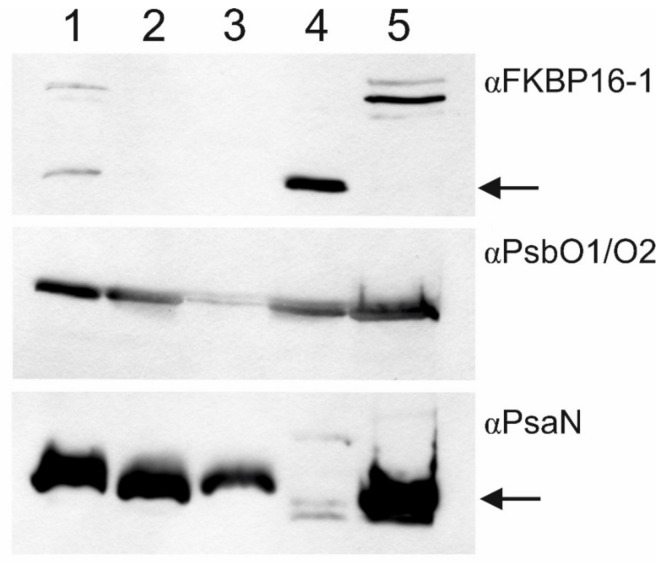
Estimation of the efficacy of free and membrane-associated lumen isolation. Western blot detection of FKBP16-1, PsbO1/PsbO2 and PsaN in thylakoid membrane and lumen fractions isolated during lumen preparations. 1—total thylakoids; 2—thylakoid membranes after removal of the free lumen fraction; 3—thylakoid membranes after removal of the membrane-associated lumen fraction; 4—free lumen; 5—membrane-associated lumen. Lanes 1–3 contain 10 µg protein, lanes 4–5 contain 30 µg protein. Arrows indicate protein of interest.

**Figure 3 ijms-22-08126-f003:**
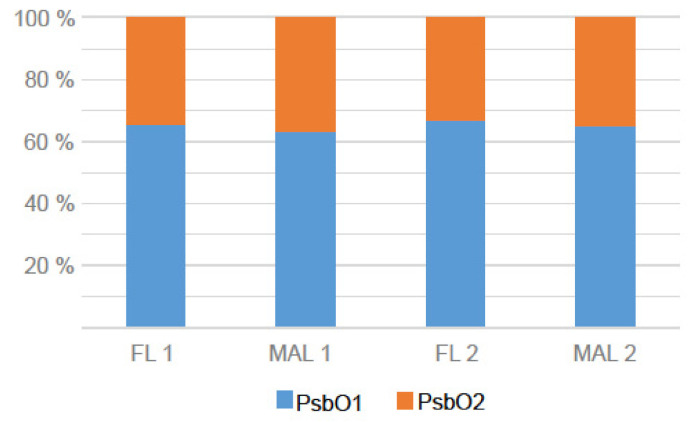
Estimation of PsbO1 and PsbO2 contents in lumen fractions. The relative abundances of PsbO1 (blue) and PsbO2 (orange) in the free lumen (FL) and membrane-associated lumen (MAL) fractions were quantified by selected reaction monitoring of their proteotypic peptides (Appendix A). Analyses of two biological replicates of each fraction are shown.

**Figure 4 ijms-22-08126-f004:**
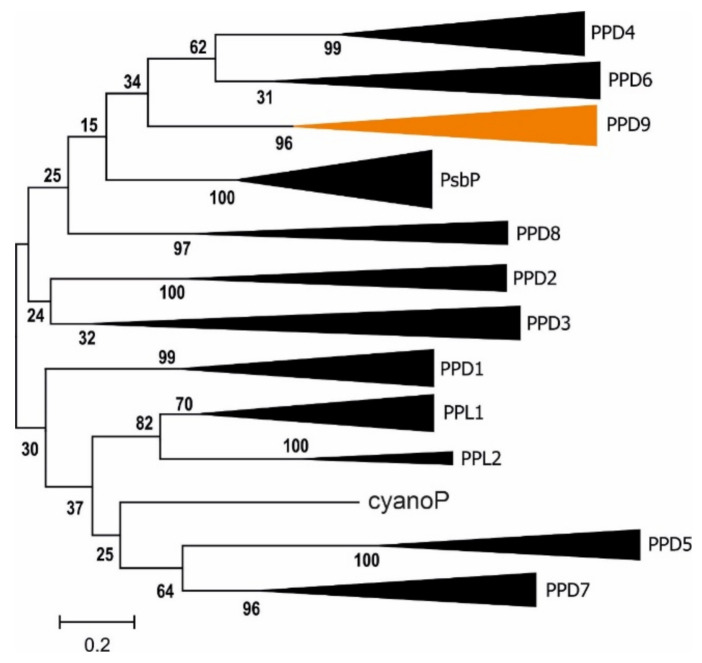
Phylogenetic analysis of PsbP families of 11 photosynthetic organisms. Compressed maximum likelihood tree shows relationships between full length putative PsbP domain-containing proteins isolated from the sequenced genomes of *Arabidopsis thaliana*, *Oryza sativa*, *Amborella trichopoda*, *Medicago truncatula*, *Physcomitrella patens*, *Marchantia polymorpha*, *Selaginella moellendorffi*, *Volvox carteri*, *Ostreococcus lucimarinus*, *Chlamydomonas reinhardtii* and *Synechocystis* sp. PCC 6803 (cyanoP; sll1418). Each node except cyanoP contains an orthologous group that is named according to homology with the corresponding PsbP domain-containing protein in Arabidopsis. The node corresponding to the novel PPD9 described here is colored in orange. Numbers show bootstrap values that indicate confidence in branching points. Scale shows substitutions per site.

**Figure 5 ijms-22-08126-f005:**
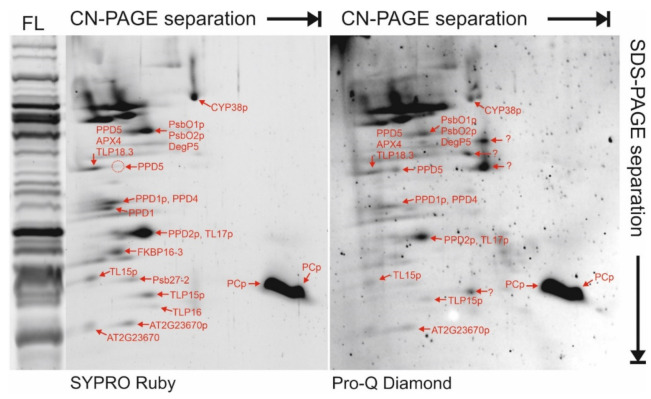
Detection of potential protein complexes and phosphorylation in the thylakoid lumen proteome. The free lumen proteome was separated in two dimensions, first by clear native (CN)-PAGE followed denaturing SDS-PAGE containing 15% acrylamide. The same gel was treated with Pro-Q Diamond phosphoprotein stain, and then with SYPRO Ruby total protein stain. Mono-dimensional separation of the same free lumen (FL) fraction on SDS-PAGE containing 15% acrylamide (stained with SYPRO Ruby) is included as a reference on the left-hand side. Protein spots of interest were excised from the 2D gel, digested with trypsin and identified using mass spectrometry. Proteins identified are indicated on the SYPRO Ruby-stained gel (left), and in some cases also on the Pro-Q Diamond-stained gel (right). The letter ‘p’ indicates that a peptide of the protein was found by MS to be phosphorylated in that spot. Note, some proteins were identified in >1 spot. See Table 2 for more details about phosphorylation. See Appendix A and Appendix A for data describing all proteins detected in each spot.

**Table 1 ijms-22-08126-t001:** Lumen proteins identified in the free lumen (FL) and membrane-associated lumen (MAL) by mass spectrometry-based proteomics.

Accession	Name *	Description	Pred. MW (Mature)	Ave. PSMs FL **	Std. Dev. PSMs FL	Ave. PSMs MAL **	Std. Dev. PSMs MAL	N Terminus ***	Pub. N Terminus ****	Proteomic Localis. Study
AT5G23120.1	HCF136 (YCF48)	Photosystem II stability/assembly factor	35.9	671.25	46.00	150.00	48.51	79	79	[16,17]
AT4G09010.1	APX4 (TLP29)	Non-functional ascorbate peroxidase	29.3	411.50	56.13	231.00	31.46	NF	83	[16,17]
AT5G66570.1	PsbO1	Photosystem II subunit O-1	26.6	280.00	27.54	342.75	51.32	NF	86	[16,17]
AT3G50820.1	PsbO2	Photosystem II subunit O-2	26.6	271.00	35.97	341.75	58.53	NF	85	[16,17]
AT1G06680.1	PsbP1	Photosystem II subunit P-1	21.0	294.75	171.22	406.50	58.16	(70) 78	78	[16,17]
AT4G21280.1	PsbQ1	Photosystem II subunit Q-1	17.0	42.00	4.55	70.00	7.26	NF	76	[16,17]
AT4G05180.1	PsbQ2	Photosystem II subunit Q-2	16.4	52.75	6.29	41.50	4.43	83 **(N-ac)**, 88	83	[17]
AT3G27925.1	DegP1	DegP protease 1	35.2	215.50	20.14	34.25	10.11	106	106	[16,17]
AT4G18370.1	DegP5	DegP protease 5	27.1	27.00	2.16	3.75	1.50	74	74	[16,17]
AT5G39830.1	DegP8	Trypsin family protein with PDZ domain	37.5	111.50	19.96	10.25	4.03	91	91	[16]
AT4G17740.1	D1 protease	Peptidase S41 family protein	41.9	60.50	10.54	13.00	4.97	127	127	[16]
AT5G46390.2	CTPA1 (D1 protease-like)	Peptidase S41 family protein	46.0	49.25	5.32	9.00	3.46	NF	68	[16]
AT3G57680.1	CTPA3	Peptidase S41 family protein	43.9	0.25 **	NA	NF **	NA	NF	none	current study, below cut-off
AT1G20340.1	PC major	Plastocyanin major form	10.5	16.75	4.99	4.25	2.22	69	69	[16,17]
AT1G76100.1	PC minor	Plastocyanin minor form	10.5	4.50	1.00	1.75	0.50	73	73	[16,17]
AT1G08550.1	VDE	Violaxanthin deepoxidase	39.8	134.50	30.96	20.00	6.38	NF	114	[16]
AT3G47860.1	LCNP (CHL)	Chloroplastic lipocalin	28.4	8.50	1.29	21.00	8.91	K_101	none	current study
AT5G64040.1	PsaN	Photosystem I reaction centre subunit PSI-N	10.6	12.75	4.27	39.50	15.84	87	85/87 (N-ac)	[17]
AT3G55330.1	PPL1	Photosystem II reaction centre PsbP family protein	17.8	111.25	24.90	39.50	14.75	NF	75	[16,17]
AT2G39470.1	PPL2	Photosystem II reaction centre PsbP family protein	18.7	0.50 **	NA	0.25 **	NA	NF	87	[17]
AT4G15510.1	PPD1	Photosystem II reaction centre PsbP family protein	21.3	109.25	14.86	12.00	6.48	NF	105	[16,17]
AT2G28605.1	PPD2	Photosystem II reaction centre PsbP family protein	14.7	22.75	1.89	4.50	3.11	**R_78**	99	
AT1G76450.1	PPD3	Photosystem II reaction centre PsbP family protein	18.8	41.75	34.38	11.50	2.38	81	81	[16,17]
AT1G77090.1	PPD4	Photosystem II reaction centre PsbP family protein	20.8	28.50	3.70	11.75	2.06	NF	72	[16,17]
AT5G11450.1	PPD5	Photosystem II reaction centre PsbP family protein	24.9	76.75	17.46	23.00	8.29	80	80	[16]
AT3G56650.1	PPD6	Photosystem II reaction centre PsbP family protein	21.6	113.25	35.17	21.50	8.39	(61) 68	68	[16,17]
AT3G05410.2	PPD7	Photosystem II reaction centre PsbP family protein	23.2	3.00	1.83	2.00	0.82	K_87	none	current study
AT5G27390.1	PPD8	Photosystem II reaction centre PsbP family protein	17.5	27.50	5.26	8.25	2.22	R_70	none	[22]
AT3G63525.1	PPD9 (TL19)	Photosystem II reaction centre PsbP family protein	20.8	76.00	15.64	13.25	4.57	**43**	none	[16], current study
AT1G14150.1	PQL1	photosystem II subunit Q-like	17.4	2.25	1.50	1.75	0.96	K_79	79	[19]
AT3G01440.1	PQL2	photosystem II subunit Q-like	13.8	2.25	1.50	1.75	0.96	**75**	126	[19]
AT2G01918.1	PQL3	photosystem II subunit Q-like	14.1	1.50	1.00	0.75 **	NA	**67**	none	current study
AT3G01480.1	CYP38	Cyclophilin-like peptidyl-prolyl cis-trans isomerase family protein	38.3	330.50	69.95	126.00	13.29	93 **(N-ac)**	93	[16,17]
AT3G15520.1	CYP37	Cyclophilin-like peptidyl-prolyl cis-trans isomerase family protein	37.9	132.75	38.59	53.00	8.12	115	115	[16]
AT5G35100.1	CYP28	Cyclophilin-like peptidyl-prolyl cis-trans isomerase family protein	28.4	92.50	11.00	18.25	3.59	20, 21, 24, 25, 30, 60	25	[16]
AT1G74070.1	CYP26-2	Cyclophilin-like peptidyl-prolyl cis-trans isomerase family protein	26.2	37.00	9.56	31.25	4.27	**75**	none	current study
AT5G13120.1	CYP20-2	Cyclophilin-like peptidyl-prolyl cis-trans isomerase family protein	20.9	83.75	31.19	33.00	3.16	**73, 74 (±N-ac)**, 77	77	[16,17]
AT5G45680.1	FKBP13	FKBP-like peptidyl-prolyl cis-trans isomerase family protein	13.6	17.75	2.99	4.75	3.59	80	80	[16]
AT4G26555.1	FKBP16-1	FKBP-like peptidyl-prolyl cis-trans isomerase family protein	15.5	16.00	5.35	2.25	0.96	**68**	none	[20]
AT4G39710.1	FKBP16-2	FKBP-like peptidyl-prolyl cis-trans isomerase family protein	15.5	9.00	2.71	1.33	0.58	**76**	94	[17]
AT2G43560.1	FKBP16-3	FKBP-like peptidyl-prolyl cis-trans isomerase family protein	15.7	25.00	2.94	13.00	4.97	77	77	[16,17]
AT3G10060.1	FKBP16-4	FKBP-like peptidyl-prolyl cis-trans isomerase family protein	14.4	16.75	3.86	17.75	2.99	R_92	95	[17]
AT4G19830.1	FKBP17-1	FKBP-like peptidyl-prolyl cis-trans isomerase family protein	16.7	3.25	0.50	0.25	0.50	**79**	none	current study
AT1G18170.1	FKBP17-2	FKBP-like peptidyl-prolyl cis-trans isomerase family protein	16.9	3.25	0.50	10.25	3.30	**29 (±N-ac), 30**, 61, **90**	61	[21]
AT1G20810.1	FKBP18	FKBP-like peptidyl-prolyl cis-trans isomerase family protein	17.8	42.00	5.77	14.00	1.83	NF	72	[16]
AT5G13410.1	FKBP19	FKBP-like peptidyl-prolyl cis-trans isomerase family protein	18.8	44.25	9.43	16.50	2.38	89	89	[16,17]
AT3G60370.1	FKBP20-2	FKBP-like peptidyl-prolyl cis-trans isomerase family protein	19.9	28.75	9.74	17.50	3.87	NF	68	[16]
AT2G44920.2	TL15 PRP	Pentatricopeptide repeat superfamily protein	15.0	53.25	13.33	25.03	7.50	82	82	[19]
AT5G53490.3	TL17 PRP	Pentatricopeptide repeat superfamily protein	18.8	215.00	26.37	22.75	12.89	NF	92	[16,17]
AT1G12250.1	TL20.3 PRP	Pentatricopeptide repeat superfamily protein	20.4	9.25	3.20	14.00	3.83	**91**	101	[19,21]
AT3G26060.1	PrxQ	Thioredoxin superfamily protein	16.7	45.00	3.92	20.50	4.51	68	68	[23]
AT5G52970.1	TLP15	Thylakoid lumen 15.0 kDa protein	16.5	29.75	6.65	3.50	2.65	76	76	[16,17]
AT4G02530.1	MPH2 (TLP16)	Thylakoid lumenal 16 kDa protein	15.7	14.50	4.89	16.33	18.35	74	74	[16,17]
AT4G24930.1	TLP17.9	Thylakoid lumenal 17.9 kDa protein	18.0	27.25	4.19	6.25	2.63	64	64	[16,17]
AT1G54780.1	TLP18.3	Thylakoid lumen 18.3 kDa protein, putative phosphatase	22.2	12.00	4.08	5.00	2.16	85	85	[16,17]
AT5G42765.1	AT5G42765	Uncharacterised protein	18.3	21.25	3.59	12.50	1.00	**65, 66, 67, 74**	86	[21]
AT2G26340.1	AT2G26340	Uncharacterised protein	20.4	11.25	3.40	11.25	5.32	81	none	current study
AT1G03600.1	Psb27-1	Photosystem II repair protein	13.6	10.75	9.18	6.50	3.79	69	69	[17]
AT1G05385.1	Psb27-2 (LPA19)	Photosystem II D1 precursor processing protein	14.9	2.25	0.96	NF **	NA	**68**	150	[21]
AT1G51400.1	PsbTn2	Photosystem II 5 kD protein	11.4	0.25 **	NA	NF **	NA	NF	none	current study, below cut-off
AT5G45040.1	Cyt c6a	Cytochrome c		3.75	2.06	0.50	1.00	**71**	none	current study
AT2G23670.1	AT2G23670	YCF37-like protein	10.0	3.50	0.58	2.75	0.96	72	72	[21]
AT2G34860.1	PSA2 PDI	Photosystem I assembly 2, DnaJ-like protein	11.9	2.75	0.50	0.75 **	NA	R_104	none	[19]
AT2G36145.1	AT2G36145	Expressed protein, predicted lumenal	12.3	0.50 **	NA	1.50	1.00	**K_62**	75	[21]
AT3G03630.1	CS26	Cysteine synthase 26	34.3	0.25 **	NA	NF **	NA	R_118	none	current study, below cut-off

* Dashed outline of cell indicates relative FL enrichment; double outline of cell indicates relative MAL enrichment; ** Indicates fewer than 2 PSMs identified in at least 2 biological replicate samples, NF (not found) indicates no PSMs found in any biological replicates, NA (not applicable) indicates that standard deviation was not calculated; *** States the first position of the N-terminal tryptic (indicated by R or K) or semi-tryptic peptide in the current study. Bolded text indicates novel N-termini or novel N terminal acetylation (N-ac) identified in this study. The number in parenthesis indicates that an N-terminal peptide upstream the mature form was found; **** The starting position of the mature protein is stated as published. If several N-termini have been published, the most N-terminal position is listed.

## Data Availability

All proteomics data have been submitted to PRIDE (dataset identifier PXD027229 and PXD027514) and Panorama (https://panoramaweb.org/fvhab3.url) repositories.

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
