# Peer review of "Characterization of the Free and Membrane-Associated Fractions of the Thylakoid Lumen Proteome in *Arabidopsis thaliana"

_ijms, 2021, doi:10.3390/ijms22158126_

Round 1

Reviewer 1 Report

This is a highly interesting work, an important addition to the series of works by E-M- Aro and coworkers to explore the identity and roles of the lumenal proteins of thylakoids.

I have only technical remarks:

  • A few of the figure legends are too short to convey the meassage
  • In Table I, describe the meaning of grey shaded entries
  • Some of the subscripts are missing (esp in M&M)

Author Response

We are grateful for the feedback from the reviewer. Please see below for details on how we have addressed the reviewer's comments.

  • A few of the figure legends are too short to convey the message

Authors' response: In the revised version, the figure legends have been separated from the text and now provide a more extensive explanation of the figures

  • In Table I, describe the meaning of grey shaded entries

Authors' response: The shaded areas of Table 1 are coloured blue, green or pink to represent FL or MAL enrichment, or novel N-termini, respectively. This is detailed in the Table legend. Hopefully the reader has access to the online version of the manuscript, as IJMS is an open access journal. The colours used in Table 1 have been adjusted in the revised version to make them more prominent.

  • Some of the subscripts are missing (esp in M&M)

Authors' response: These mistakes have been corrected. We apologise for this oversight.

Reviewer 2 Report

The manuscript by Gollan et al. reports proteome profiles of soluble and membrane-associated lumen fractions of thylakoid membranes from Arabidopsis chloroplasts. Their data include several new findings about the nature of lumenal proteins and provide very important datasets available for future detailed analyses of individual proteins of known and unknown functions in the lumen. The manuscript is written in a concise and straightforward manner and I have only some minor comments on their manuscript.

  1. The legends for figures excluding Fig. 1 appear in the same body as the main text and are difficult to be recognized. Please correct this point.
  2. The sentence in line 109 may be "Western blot detection of FKBP16-1... and PsaN [in] thylakoid membrane and lumen fractions isolated during lumen preparations."
  3. I cannot understand the meanings of the following notes "** Indicates fewer than 2 PSMs in Figure 2. biological replicate samples identified" and "*** The starting position of N-terminal tryptic semi-tryptic or tryptic peptide in the current study is stated" for Table 1 in the lines 140-142.
  4. Throughout the manuscript, scientific names of species will be written in italic.
  5. In lines 423-426, the authors discussed the possibility of the existence of acetyl transferase and acetyl CoA in the lumen. In addition, are there any possibilities of N-terminal acetylation before the transport of such proteins into the lumen, as discussed for the case of phosphorylation in the stroma?
  6. I cannot find the legends for Supplementary Figure S1 and S2, so it is hard to understand the data. Please include them in the supplementary files.

Author Response

We are grateful for the reviewer's feedback. Please see below for our responses to the reviewer's comments.

1. The legends for figures excluding Fig. 1 appear in the same body as the main text and are difficult to be recognized. Please correct this point.

Authors' response: Thank you for pointing out this error. It has been corrected in all Figures in the revised version

2. The sentence in line 109 may be "Western blot detection of FKBP16-1... and PsaN [in] thylakoid membrane and lumen fractions isolated during lumen preparations."

Authors' response: This has been corrected, see line 109 in the revised manuscript

Reviewer 3 Report

In this manuscript, the authors conducted proteomic studies on thylakoid lumen proteins in vegetative tissues of Arabisopsis thaliana. They identified various proteins in the soluble free lumen (FL) and the membrane-associated lumen (MAL) fractions, and showed the difference in the protein compositions between FL and MAL. They also showed the new PsbP domain-containing proteins, and identified proteins that have far escaped N-terminal analysis. Most of data were sound and their description is precise. Only the following minor flaws should be improved.

1) In L. 96, they should mention what protein is FKBP16-1 for unfamiliar readers.

2) In L. 194, “Chlamydomonas reinhardtii” should be changed to italic form. Other species names should be also changed.

3) In this study, the authors reported the clear proteomic difference between FL and MAL. Can they show these differences as a simple illustration for unfamiliar readers.

Author Response

We are grateful for the reviewer's feedback. Please see below for our responses to the reviewer's comments.

1. In L. 96, they should mention what protein is FKBP16-1 for unfamiliar readers.

Author's response: We thanks the reviewer for this advice. The phrase “FK506-binding protein” has been inserted in the revised version in lines 95-96

2. In L. 194, “Chlamydomonas reinhardtii” should be changed to italic form. Other species names should be also changed.

Authors' response: The missing italics have been introduced throughout the manuscript

3. In this study, the authors reported the clear proteomic difference between FL and MAL. Can they show these differences as a simple illustration for unfamiliar readers.

Authors' response: We thank the reviewer for this suggestion. A graphical abstract has been included to explain this concept in a simple way